# Sexual harassment before and during the COVID-19 pandemic among adolescent girls and young women (AGYW) in Nairobi, Kenya: a cross-sectional study

Kristin G Bevilacqua [1], A Williams,[1] Shannon N Wood [1,2] G Wamue-Ngare,[3,4] Mary Thiongo,[5] P Gichangi,[5,6] Michele R Decker[1,2]

For numbered affiliations see end of article.

**Correspondence to**
Dr Michele R Decker;
mdecker@jhu.edu

## ABSTRACT

**Objectives** Sexual harassment among adolescent girls and young women (AGYW) is a prevalent and understudied form of gender-based violence (GBV) with negative impacts on health and well-being. The COVID-19 pandemic raised global concern about GBV within homes; less is known about how it affected GBV in public spaces.

**Methods** Present analyses use cross-sectional data from a cohort of adolescents and young adults residing in Nairobi, Kenya, restricted to female participants. Data were collected August–October 2020 via phone after implementation of COVID-19 restrictions. Prevalence of past-year sexual harassment and harassment relative to COVID-19 restrictions were calculated for overall sample, and by individual, household, and pandemic-related factors. Multivariate negative binomial regression models examine correlates of (1) past-year sexual harassment and (2) increases in sexual harassment relative to COVID-19 restrictions.

**Results** Overall, 18.1% of AGYW experienced past-year sexual harassment at the 2020 survey. Among this group, 14.6% experienced sexual harassment pre-COVID-19 only, 18.8% after only and 66.6% at both time points. Among the latter group, 34.9% reported more occurrences following COVID-19 restrictions, 20.5% reported less occurrences and 44.7% reported no change in occurrence. Overall, 42.0% of AGYW experienced an increase in sexual harassment while 58.0% experienced no increase since COVID-19. In adjusted models, past-year sexual harassment was associated with higher educational attainment (adjusted risk ratio, aRR 2.11; 95% CI 1.27 to 3.52) and inability to meet basic financial needs (aRR 1.67; 95% CI 1.05 to 2.66). Increased sexual harassment since COVID-19 was associated with having full control to leave the home (aRR 1.69; 95% CI 1.00 to 2.90).

**Conclusions** Sexual harassment among AGYW in Nairobi, Kenya was prevalent before and during COVID-19 restrictions. Safety in public spaces remains a highly gendered issue that impacts women's safety and ability to participate in public life. Prevention and support services to address sexual harassment remain an important element in ensuring safe, sustainable public spaces.

## STRENGTHS AND LIMITATIONS OF THIS STUDY

⇒ This is the first study to examine prevalence of sexual harassment in relation to the COVID-19 pandemic among adolescent girls and young women (AGYW) in Nairobi, Kenya.

⇒ We focus on AGYW, a key developmental time period during which experiences of violence may have long-term consequences for health, well-being, and future risk of gender-based violence.

⇒ Cross-sectional, retrospective data limit our ability to explore causal relationships.

⇒ Use of a single-item outcome measure may have reduced sensitivity, increasing risk of misclassification (ie, under-reporting) of sexual harassment.

## BACKGROUND

Global awareness of sexual harassment has risen steadily since 2017, stemming from the global #MeToo, Times Up and Generation Equality movements. Yet relative to other leading forms of gender-based violence (GBV), sexual harassment remains understudied. Sexual harassment can include staring, cat calling, forced conversation and/or stalking by either strangers, peers, acquaintances or authority figures, and often occurs in public spaces, such as on the street or in workplace or educational settings, but can also occur in the home and other private spaces.[1] A recent systematic review meta-analysis found wide-ranging prevalence estimates from 0.6% to 26.1%, globally,[2] potentially reflecting differences in the conceptualisation and measurement of sexual harassment, which occurs across diverse spaces and perpetrators.[3–5] Sexual harassment can serve as a potent reminder of underlying gender imbalances and predatory sexual climates known as coercive sexual environments,[6] which

disproportionately burden adolescent girls and young women (AGYW).[5 7 8]

Experiences of sexual harassment during adolescence and young adulthood can have serious consequences for the health and well-being of AGYW. Sexual harassment contributes to increased risk of work and school absenteeism, compromising AGYW's future economic stability.[9 10] For example, harassment experienced on public transportation may compromise AGYW's labour or educational participation and advancement if they are unable to travel safely.[7 11] Experiences of sexual harassment are also associated with a host of negative mental health outcomes, including stress, anxiety, and fear.[10 12 13] Further, sexual harassment among AGYW is associated with elevated risk of future harassment, as well as physical and sexual violence victimisation,[14] reflecting embedded social/environmental risks. Despite its consequences, there is a dearth of research aimed at understanding risk factors for sexual harassment among AGYW,[4] and existing research is largely concentrated in high-income countries or among high-income subgroups within low-income and middle-income countries (LMICs).[2]

Mirroring gaps in sexual harassment research in LMICs, a few studies have explored sexual harassment specifically within Kenya.[15 16] The limited available literature is qualitative and offers useful insight into sexual harassment among AGYW. In one case study at University of Nairobi, women described sexual harassment as a main barrier to feeling comfortable taking certain classes, using university toilets, and accessing certain spaces on campus, such as the library.[15] Another study among high school AGYW in Nairobi's urban settlements highlighted how sexual harassment experienced in transit to and from school often escalated from verbal harassment to unwanted touching, and when men's attention was not reciprocated, it could lead to violence.[17] Though quantitative assessments of sexual harassment among AGYW in Kenya are not currently available, qualitative findings align with experiences of sexual harassment among this population in other settings and highlight the gendered risks AGYW face across public contexts such as school, work, and public transportation.[18 19]

The public nature of sexual harassment distinguishes it from other forms of GBV like intimate partner violence, which occur more often in private spaces. The COVID-19 pandemic raised global concern for GBV in private spaces due to increased time in the home with potentially abusive partners;[20] however, the potential impacts of changes to AGYW's mobility and time spent outside the home on experiences of GBV are less clear. For example, closures and mobility restrictions due to pandemic policies may decrease sexual harassment risk in some settings, such as schools, but could increase risk of sexual harassment among AGYW travelling in spaces made more isolated by pandemic restrictions. Early evidence of these dynamics has been observed in Kenya; a 2021 UN report on violence against women during the pandemic showed that 55% of women in Kenya reported that COVID-19 made them feel less safe walking alone at night and 81% perceived an increase in sexual harassment since the start of the pandemic.[21] However, to our knowledge, no studies have explored the prevalence of sexual harassment or risk factors associated with potential COVID-19-related increases among the priority population of AGYW in Kenya.

The United Nations' Sustainable Development Goal 11 calls for the promotion of safe, resilient and sustainable cities and settlements.[22] Efforts to achieve this goal must employ a gendered perspective to better understand and address the violence and hostility experienced by girls and women in public spaces.[7] To this end, this study uses cross-sectional data from a cohort of youth in Nairobi, Kenya to examine (1) the prevalence of past-year sexual harassment among a sample of AGYW overall, and relative to the implementation of COVID-19 restrictions, (2) factors associated with any past-year sexual harassment and (3) factors associated with increased sexual harassment relative to COVID-19 restrictions.

## METHODS
### Study population
The study population is a subsample drawn from the Nairobi Youth Respondent Driven Sampling (RDS) Survey, an ongoing cohort study of adolescents and young adults first recruited prepandemic, in 2019. Eligible participants were age 15–24 years, unmarried, and residing in Nairobi for at least 1 year at the time of enrolment. Using RDS methods, seeds (n=9) were purposefully selected to catalyze peer-to-peer recruitment via coupon distribution (up to three per person) until the target sample size was achieved. In August 2019, 1357 young men and young women were surveyed, 95% of whom (1,293/1,357) consented for recontact. At 12-month follow-up (August–October 2020), 1217 (94%) were successfully recontacted and consented and completed the follow-up survey. Additional sampling and recruitment details can be found elsewhere.[23] Sexual harassment experiences and their timing relative to COVID-19 pandemic restrictions were assessed at the 2020 survey wave only; the present analytic sample is restricted to all female participants in the 2020 survey (n=612).

### Data collection
COVID-19 restrictions were first implemented in Kenya in March 2020 when President Kenyatta ordered the closing of schools and non-essential workplaces and barred travel from countries with reported cases of COVID-19, among other restrictions.[24] Due to the implementation of COVID-19 restrictions, trained resident enumerators (REs) conducted data collection by phone in either English or Swahili. Oral consent was recorded electronically via OpenDataKit. The Ethics Review Committee (ERC) at Kenyatta National Hospital/University of Nairobi and the Institutional Review Board at Johns Hopkins Bloomberg School of Public Health

waived parental consent for this study and approved use of oral consent processes. All consent procedures aligned with ethical best practices for sensitive topics, including specialised training, privacy protections (auditory privacy screener and protocol), and provision of resource referrals. Participants who consented for recontact in 2019 were consented prior to the 2020 follow-up. All data were collected in accordance with best practices for GBV research, and RE's received specialised training specific to GBV protections.[25] To ensure privacy and safety for GBV and other sensitive topics during remote data collection, RE's confirmed participant safety and privacy and rescheduled as needed. GBV support services were provided to all participants within a larger list of supports to minimise risk. Participants received KES500 or US$5 per survey completed.

## Measures
### Primary outcomes
Past-year sexual harassment was assessed with a single item adapted from seminal research on coercive sexual environments[6]: 'In the past 12 months, have you experienced unwanted sexual attention or harassment such as verbal comments, staring or leering, or unwanted physical contact like groping or grabbing?' Response options included: never, once, a few times, often or no response. Among AGYW who reported any past-year sexual harassment, participants were asked to report timing relative to the COVID-19 restrictions (March 2020); 'Have experiences of unwanted sexual attention or harassment happened before COVID-19 restrictions only, since COVID-19 restrictions only, or both?' AGYW who reported sexual harassment both before and after COVID-19 restrictions described changes in occurrences; 'Have experiences of sexual harassment changed since the COVID-19 restrictions began?' Response options: increased a lot, increased somewhat, no change, decreased somewhat, decreased a lot. To maximise power for analyses, increased sexual harassment since the pandemic, among those who experienced past year sexual harassment, was modelled dichotomously: AGYW were classified as having increased sexual harassment if they first started experiencing sexual harassment after pandemic restrictions began or if they experienced sexual harassment both before and after pandemic restrictions but reported an increase in occurrence since the start of the pandemic. AGYW were classified as no increase in sexual harassment if they experienced sexual harassment only before the pandemic or if they experienced sexual harassment before and after pandemic restrictions but reported a decrease in occurrence during the pandemic or if they experienced sexual harassment before and after pandemic restrictions but reported no change in occurrence relative to the pandemic.

### Independent variables
Individual, household, and pandemic-related covariates were measured in 2020. At the individual level, demographic assessments including age, educational attainment, current school enrolment status, and marital status were obtained. AGYW's control over the decision to leave the house and over their own earnings was assessed as part of a broader measure of personal agency via 4-point Likert scale, with categories collapsed for analysis (0='less than full control,' or 1='full control').[26] Household-level covariates were composed of subjective household socioeconomic status (SES)[27] and household composition (response options: 0= 'living alone,' 1= 'living with parents with or without others,' 2= 'living with partner with or without others,' 3= 'living with others'). The pandemic-related covariate focused on inability to meet basic needs since the start of the COVID-19 pandemic, assessed via 4-point Likert scale and dichotomised for analysis (0= 'basic financial needs met,' or 1= 'basic financial needs not met').

## Analysis
We first calculated the proportion of AGYW who experienced sexual harassment in the 12 months preceding the survey and the timing of these experiences relative to the COVID-19 restrictions (only before, only after, or both). Among those who reported sexual harassment both before and after pandemic restriction initiation, we characterised changes in occurrences relative to COVID-19 restrictions (ie, no change, decreased occurrence, or increased occurrence) and overall changes in sexual harassment since COVID-19 restrictions (ie, increase or no increase). We examined each past-year sexual harassment and overall increased sexual harassment since COVID-19 restrictions by demographics, with differences assessed via design-based F-statistics. Sexual harassment relative to COVID-19 pandemic restrictions among AGYW who experienced past-year sexual harassment (n=118) were visualised using a Sankey diagram (figure 1).

Based on unadjusted bivariate associations, multivariate negative binomial regression models were constructed to examine correlates of (1) past-year sexual harassment and (2) increases in sexual harassment relative to COVID-19 restrictions, respectively. All analyses were conducted using Stata V.15.1 with statistical significance set a priori at $p < 0.05$; threshold for non-significant trend set at $p < 0.10$. Sampling weights accommodate the RDS study design using RDS-II (Volz-Heckathorn) weights, and modest adjustments for postestimation and loss to follow-up. Statistical testing accounts for survey weights and complex survey design.

## Patient and public involvement
This community-engaged study incorporated community input throughout its inception, implementation, and dissemination. During its inception, key stakeholders from community-based, youth-serving organisations consulted on study recruitment, survey design, and data collection procedures. Trained REs were recruited from study communities and provided integral input on survey

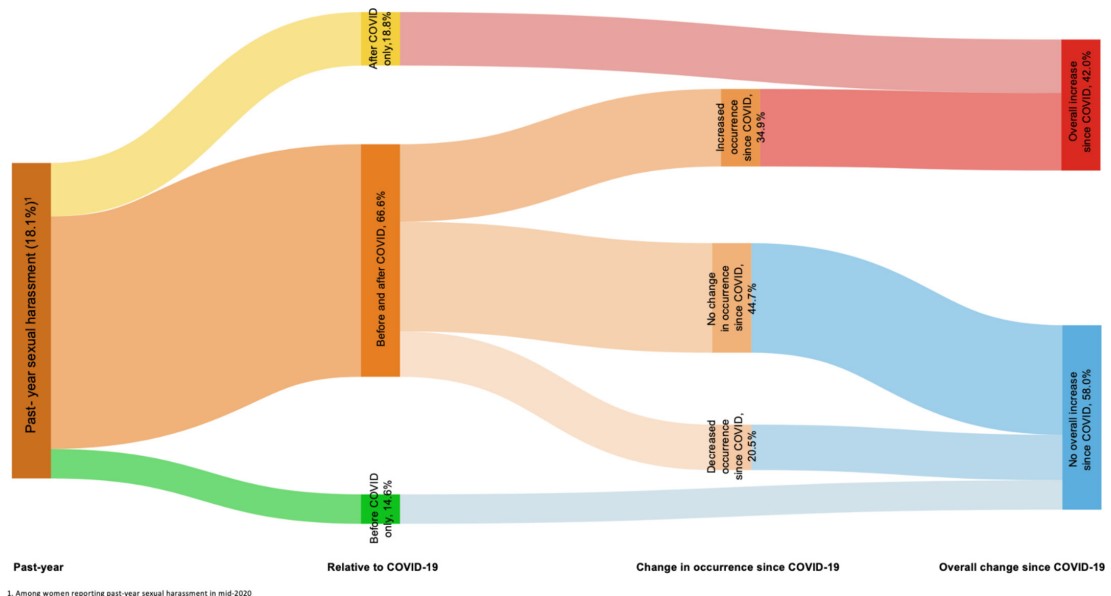

**Figure 1** Trajectories of sexual harassment relative to and changes in frequency since the COVID-19 pandemic among adolescent girls and young women in Nairobi, Kenya in mid-2020 (n=612).

design and data collection. Findings have been disseminated among key stakeholders, including government representatives, community and faith leaders, community-based organisations, and youth leaders.

## RESULTS

Overall, 18.1% of AGYW reported past-year sexual harassment (table 1). Among those who experienced past-year sexual harassment, 66.6% experienced harassment both before and after COVID-19 restrictions (March 2020), 14.6% experienced harassment only before and 18.8% experienced harassment only after COVID-19 restrictions. Among those endorsing sexual harassment both before and after COVID-19 restrictions, 44.7% experienced no change in occurrence, 34.9% experienced an increase in occurrence and 20.5% experienced a decrease in occurrence. Among AGWY who experienced any past-year sexual harassment, 42.0% experienced an overall increase relative to COVID-19 (ie, new or increased sexual harassment since March 2020) (figure 1).

### Correlates of past-year sexual harassment

In bivariate analyses, past-year sexual harassment was more prevalent among AGYW with greater educational attainment (college/university vs secondary or lower; 38.5% vs 15.9%; p<0.01) (table 2), whereas not being enrolled in school at the time of the survey was marginally associated with past-year sexual harassment (20.9% vs 14.2%; p=0.11). AGYW who were unable to meet their basic financial needs since COVID-19 were more likely to experience past-year sexual harassment than those able to meet their basic financial needs (22.6% vs 12.9%; p=0.02). In adjusted multivariate negative binomial models, inability to meet basic financial needs remained significantly associated with past-year sexual harassment

(adjusted risk ratio, aRR 1.67, 95% CI 1.05 to 2.66), as did higher educational attainment (aRR 2.11, 95% CI 1.27 to 3.52) (table 2).

**Table 1** Sexual harassment prevalence, timing and changes relative to COVID-19 pandemic restrictions among AGYW, 2020 (n=612)

| | Weighted % (n) |
| --- | --- |
| Experienced in past 12 months | 18.1 (118) |
| Timing relative to COVID-19 restrictions; among those who experienced past-year sexual harassment (n=118) | |
| Pre-COVID-19 only | 14.6 (11) |
| Post- COVID-19 only | 18.8 (20) |
| Both time periods | 66.6 (87) |
| Changes in occurrence since COVID-19 restrictions; among those who experienced sexual harassment both prior to and during COVID-19 restrictions (n=87) | |
| No change | 44.7 (41) |
| Decreased occurrence | 20.5 (17) |
| Increased occurrence | 34.9 (29) |
| Overall increase in sexual harassment since COVID-19 restrictions began, among those who experienced past-year sexual harassment* (n=118) | |
| Increase | 42.0 (49) |
| No increase | 58.0 (69) |

*New or increased sexual harassment since March 2020.
AGYW, adolescent girls and young women.

**Table 2**  Factors associated with past-year sexual harassment among AGYW, weighted

| | Sample (n=612) | Past-year sexual harassment | | |
|---|---|---|---|---|
| | Col% | Row% | P value† | aRR (95% CI)‡ |
| Overall | – | 18.1 | – | – |
| Age group | | | 0.68 | |
| 16–20 years | 42.6 | 19.1 | | – |
| 20–26 years | 57.4 | 17.3 | | – |
| Highest level of education completed | | | <0.01** | |
| Secondary or lower | 90.4 | 15.9 | | ref |
| College/university | 9.6 | 38.5 | | 2.11** (1.27 to 3.52) |
| Currently in school | | | 0.11 | |
| No | 58.2 | 20.9 | | ref |
| Yes | 41.8 | 14.2 | | 0.82 (0.48 to 1.40) |
| Marital status | | | 0.46 | |
| Unmarried | 90.2 | 17.6 | | – |
| Married | 9.8 | 22.6 | | – |
| Personal control over decision to leave house | | | 0.85 | |
| Less than full control | 60.0 | 18.5 | | – |
| Full control | 40.0 | 17.6 | | – |
| Control over own earnings | | | 0.29 | |
| Less than full control | 71.7 | 16.7 | | – |
| Full control | 28.3 | 21.7 | | – |
| Relative household SES tertile | | | 0.41 | |
| Lowest | 37.0 | 14.4 | | – |
| Middle | 22.2 | 19.3 | | – |
| Highest | 40.7 | 20.8 | | – |
| Household composition | | | 0.17 | |
| Live alone | 6.3 | 20.0 | | – |
| Lives with parent(s), with or without other(s) | 66.4 | 17.2 | | – |
| Lives with partner with or without other(s), excluding parent(s) | 10.0 | 30.7 | | – |
| Live with others | 17.2 | 13.4 | | – |
| Ability to meet basic financial needs since COVID-19 | | | 0.02* | |
| Basic financial needs met | 46.6 | 12.9 | | ref |
| Basic financial needs unmet | 53.4 | 22.6 | | 1.67* (1.05 to 2.66) |

*p<0.05, **p<0.01.
†P value for the design-based F statistic bivariate testing.
‡aRR generated through multivariate negative binomial regression with past-year sexual harassment as the dependent variable, among full sample (n=612); all variables listed included in specification accounting for robust SE clustering by node; weighted.
AGYW, adolescent girls and young women; aRR, adjusted risk ratio; SES, socioeconomic status.

### Correlates of overall increase in sexual harassment since COVID-19

Among AGYW who experienced any past-year sexual harassment, in bivariate analyses, those currently enrolled in school were more likely to experience increased sexual harassment since the start of the COVID-19 restrictions compared with those not currently enrolled in school (59.0% vs 33.7%; p=0.05).

Having full control over one's decision to leave the home was marginally bivariately associated with increased sexual harassment (57.4% vs 32.2%; p=0.06) (table 3). In adjusted multivariate negative binomial models, full control to leave the home was associated with overall increased sexual harassment since the start of COVID-19 restrictions (aRR 1.69; 95% CI 1.00 to 2.90) (table 3).

**Table 3** Factors associated with increased sexual harassment since COVID-19 restrictions, among AGYW with past-year sexual harassment, weighted

| | Sample (n=118) | Increased sexual harassment since covid | | |
| --- | --- | --- | --- | --- |
| | Col% | Row% | P value† | aRR (95% CI)‡ |
| Overall | – | 42.0 | – | – |
| Age group | | | 0.38 | |
| 16–20 years | 45.0 | 48.3 | | – |
| 20–26 years | 55.0 | 36.8 | | – |
| Highest level of education completed | | | 0.60 | |
| Secondary or lower | 79.5 | 40.4 | | – |
| College/university | 20.5 | 48.1 | | – |
| Currently in school | | | 0.05* | |
| No | 67.3 | 33.7 | | ref |
| Yes | 32.7 | 59.0 | | 1.66 (0.98, 2.80) |
| Marital status | | | 0.66 | |
| Unmarried | 87.8 | 43.0 | | – |
| Married | 12.2 | 35.1 | | – |
| Personal control over decision to leave house | | | 0.06 | |
| Less than full control | 61.2 | 32.2 | | ref |
| Full control | 38.8 | 57.4 | | 1.69* (1.00 to 2.90) |
| Control over own earnings | | | 0.26 | |
| Less than full control | 66.2 | 46.5 | | – |
| Full control | 33.8 | 33.2 | | – |
| Relative household SES tertile | | | 0.97 | |
| Lowest | 29.6 | 43.3 | | – |
| Middle | 23.6 | 43.3 | | – |
| Highest | 46.8 | 40.5 | | – |
| Household composition | | | 0.17 | |
| Live alone | 7.0 | 16.6 | | – |
| Lives with parent(s), with or without other(s) | 63.3 | 44.4 | | – |
| Lives with partner with or without other(s), excluding parent(s) | 17.0 | 25.9 | | – |
| Live with others | 12.8 | 65.1 | | – |
| Ability to meet basic financial needs since COVID-19 | | | 0.59 | |
| Basic financial needs met | 33.3 | 37.2 | | – |
| Basic financial needs unmet | 66.7 | 44.4 | | – |

*p<0.05.
†P value for the design-based F statistic bivariate testing.
‡aRR generated through multivariate negative binomial regression with increased sexual harassment since COVID-19 restrictions as the dependent variable, among those who experienced past-year sexual harassment (n=118); all variables listed included in specification accounting for robust SE clustering by node; weighted.
AGYW, adolescent girls and young women; aRR, adjusted risk ratio; SES, socioeconomic status.

## DISCUSSION

Sexual harassment was pervasive among AGYW in our sample, with 18.1% reporting experiences of sexual harassment in the past year. Among AGYW experiencing past-year sexual harassment, the majority reported those experiences occurring both before and after the start of COVID-19 pandemic restrictions. Within the subset of AGYW who experienced sexual harassment before and after COVID-19 restrictions were implemented, most (44.7%) experienced no change in occurrence of sexual harassment after the start of the pandemic. Overall, 42.0% experienced an increase in sexual harassment

since the start of COVID-19 pandemic restrictions. The high prevalence of sexual harassment prior to the start of the pandemic and sustained or increased occurrence of such violence echoes the burdens of other forms of GBV in Kenya[28–31] and underscores the need for attention to sexual harassment and its impacts on health and well-being among AGYW in this setting.

Past-year sexual harassment was associated with AGYW's inability to meet their basic financial needs. These results support a growing body of research that has found lower SES to be associated with increased risk of sexual harassment,[32] including during adolescence.[33–36] Interestingly, past-year sexual harassment was also associated with greater educational attainment, perhaps due to sexual harassment experienced in the school/university[37–40] or workplace settings.[41 42] We note that AGYW who were not currently enrolled in school during the COVID-19 pandemic were marginally significantly more likely to report past-year sexual harassment. These results suggest that young women who left school, potentially due to the pandemic, and more educated women who have already entered the workforce may have heightened risk. Results reflect the ways in which both lower SES and higher educational attainment may contribute to increased time in public spaces, either out of financial necessity or workplace opportunities afforded by an advanced degree, and suggest that greater access to educational and economic resources may not be protective against sexual harassment. Further, the increased risk associated with school enrolment suggests that initiatives to keep AGYW in school may not protect them from sexual harassment and that further intervention to reduce perpetration in school settings is urgently needed. Adolescence and young adulthood mark pivotal developmental junctures, during which educational and economic opportunities shape future health and well-being. While public spaces remain unsafe for AGYW, their ability to participate in education or work will continue to be compromised, compounding their socioeconomic vulnerability and risk of future violence.

A growing body of literature has documented the ways in which pandemic-related restrictions and psychosocial stressors have contributed to increased GBV risk globally.[20 28 32] In contrast to more private forms of GBV like IPV, sexual harassment is often perpetrated in public spaces, making the potential impact of COVID-19-related restrictions and stressors less clear. The majority of AGYW who had ever experienced sexual harassment did so both before and after the start of the pandemic, suggesting that for most AGYW, pandemic-related restrictions had minimal impact on their risk of sexual harassment, perhaps reflecting economic realities that prevented many Kenyans from remaining home despite stay-at-home orders.[43] In adjusted models, personal control to leave the home was significantly associated with overall increased sexual harassment since the start of the pandemic among AGYW, and current school enrolment was marginally associated with increased sexual harassment. These results

underscore the ways in which mobility restrictions may have presented a double-edged sword for AGYW, whereby remaining at home may have placed them at greater risk of IPV or violence perpetrated by family members,[28 44–46] and leaving the home may have made them more vulnerable to sexual harassment in public spaces like work and transit.

Unfortunately, the current policy landscape does not reflect the realities of AGYW at risk for sexual harassment. The Sexual Offences Act, passed by the Government of Kenya in 2006, defines sexual harassment as persistent sexual advances or requests made by any person in a position of authority or holding public office.[47] The law's narrow definition fails to address harassment perpetrated by peers or coworkers, in informal work settings, or in public. Notably, penal code 144(3), the part of the Act which prohibited insult to women's modesty or utterances of words or sounds and gestures that, 'intrude on the privacy of the woman or girl,'[47] but fell short of labelling such behaviour as sexual harassment, has since been repealed.[48] While current policies fall short, public advocacy efforts and attention surrounding sexual harassment are mounting. For example, when a young woman was harassed and assaulted by boda boda (motorcycle) drivers in March 2022, large public protests erupted.[49] Increased research is urgently needed to inform policy and intervention to address the unique aspects of sexual harassment experienced during adolescence and young adulthood, including age-specific risk factors for perpetration and victimisation, and health impacts.

Results must be understood in the context of the study's limitations. Cross-sectional, retrospective data were obtained at a single point in time and limits our ability to explore causal relationships. Surveys were conducted via mobile phone, which may have resulted in a study sample of more well-resourced youth; however, postestimation weights improve generalisability to the overall youth population in Nairobi. To minimise participant fatigue, a single item assessed sexual harassment, which may be less sensitive,[50] increasing risk of misclassification (ie, underreporting). For example, one study that measured more specific forms of sexual harassment in a similar setting found higher prevalence of such violence both before and after the COVID-19 pandemic.[32] While the parent study included a qualitative component, it was not focused on sexual harassment, which may have helped contextualise findings. Finally, due to survey space constraints, we lack details on sexual harassment context (eg, work, school, public, transit, home), perpetrators (eg, teacher, supervisor, stranger, family member) and severity (eg, level of threat, aggression and harm incurred). We note it is possible that the unwanted sexual attention occurred in the home, and our measure is not exclusive of partner harassment. We were also unable to assess how students were attending school (ie, in person or virtually), which may have influenced the sexual harassment dynamics.

Despite these limitations, taken together, results suggest an underlying gendered risk context in diverse spaces,[5 7 36]

which contributes to endemic GBV and permeates the lives of AGYW. Experiences of sexual harassment across public spaces (ie, transit, school, and work) are a reflection of patriarchal social norms that govern where and when women can occupy space[8] and the consequences for entering a 'contested'[5] space. Other scholars have characterised environments of unwanted sexual attention and pressure as 'coercive sexual environments' that are harmful in their own right, and simultaneously give rise to more egregious forms of GBV while reinforcing norms of tolerance, impunity, and victim-blaming for such experiences.[6] 'Safety work,'[5] or behaviours employed by women to avoid sexual harassment, such as limiting mobility,[11] prevent women and girls' full participation in society and can have long-term consequences for AGYW's educational attainment and economic opportunity and, in turn, their future health and well-being.[9 10] These consequences may be particularly acute for low-resourced AGYW or AGYW living in low-income environments.[36]

The momentum gained from increased attention to GBV in Kenya during the COVID-19 pandemic must be harnessed to concurrently increase focus on the burden of sexual harassment among AGYW in this setting.[51] As the severity of the COVID-19 pandemic and associated restrictions continue to fluctuate, increased funding for the development and evaluation of interventions to prevent and address sexual harassment are needed. In an important first step, President Kenyatta has pledged significant funds to address GBV in Kenya[52] and the Government of Kenya has released a National Action plan for eliminating GBV in Kenya by 2026.[53] However, these efforts focus primarily on sexual harassment in the workplace[54] and should be expanded to include sexual harassment at school, on transit, and in other public spaces. Further, future interventions must focus on sexual harassment and its sequelae specifically among AGYW to mitigate the long-term impacts of sexual harassment on the social, economic, and physical well-being of this population. Finally, efforts must focus on the norms that perpetuate the gendered risk of violence and the potential for social norms change to reduce GBV perpetration and promote help-seeking.

**Author affiliations**
¹Department of Population, Family and Reproductive Health, Johns Hopkins University Bloomberg School of Public Health, Baltimore, Maryland, USA
²Department of Population, Family and Reproductive Health, Johns Hopkins University Bloomberg School of Public Health, Bill & Melinda Gates Institute for Population and Reproductive Health, Baltimore, Maryland, USA
³Department of Sociology, Gender and Development Studies, Kenyatta University, Nairobi, Kenya
⁴Women's Economic Empowerment Hub, Kenyatta University, Nairobi, Kenya
⁵International Centre for Reproductive Health Kenya, Mombasa, Kenya
⁶Department of Public Health and Primary Care, Ghent University, Gent, Belgium

**Contributors** MRD, PG and MT conceptualised and designed the present study. Protocol development, including measures and safety protocol, was completed by MRD, PG, MT and GW-N. Analysis was conducted by AW and KGB. Manuscript preparation and writing were completed by KGB and AW. GW-N, PG, SNW and MRD edited and interpreted the data. MRD is the overall guarantor for this work.

**Funding** This work was supported by the Bill & Melinda Gates Foundation (010481). Under the Foundation grant conditions, a Creative Commons Attribution 4.0 Generic Licence has already been assigned to the author accepted manuscript version that might arise from this submission.

**Competing interests** None declared.

**Patient and public involvement** Patients and/or the public were involved in the design, or conduct, or reporting, or dissemination plans of this research. Refer to the Methods section for further details.

**Patient consent for publication** Not applicable.

**Ethics approval** This study involves human participants and procedures were approved by the Ethics Review Committee at Kenyatta National Hospital/University of Nairobi (P310/06/2020) and the Institutional Review Boards at Johns Hopkins Bloomberg School of Public Health (IRB 00012952). Participants gave informed consent to participate in the study before taking part.

**Provenance and peer review** Not commissioned; externally peer reviewed.

**Data availability statement** Data are available on request from pmadata.org.

**ORCID iDs**
Kristin G Bevilacqua http://orcid.org/0000-0002-3260-3733
Shannon N Wood http://orcid.org/0000-0003-4389-3526

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
