## [Reviewer comments · BMJ Open]

ARTICLE DETAILS

TITLE (PROVISIONAL)	Sexual harassment before and during the COVID-19 pandemic among adolescent girls and young women (AGYW) in Nairobi, Kenya: a cross-sectional study
AUTHORS	Bevilacqua, Kristin; Williams, A; Wood, Shannon; Wamue-Ngare, G; Thiongo, Mary; Gichangi, P; Decker, Michele

VERSION 1 – REVIEW

REVIEWER	N Kumar Maharishi Markandeshwar Institute of Medical Sciences and Research
REVIEW RETURNED	01-Aug-2022

GENERAL COMMENTS	The manuscript is very well written and has tried to address one of the major areas especially during the time of COVID-19 Pandemic. There are minor suggestions which if added can further increase the value of your manuscript. The reviewer provided a marked copy with additional comments. Please contact the publisher for full details.
---

REVIEWER	David Olawade University of Ibadan, Department of Environmental Health Science
REVIEW RETURNED	04-Sep-2022

GENERAL COMMENTS	This paper is very concise, comprehensive and articulate. Good knowledge of the study background and sound methodology. The findings are intriguing and the recommendations are insightful, however, there is a need for minor revision as it concerns the discussion section. There are similar cross-sectional studies carried out in other part of Africa and the world at large that the findings in this study can be compared with. Do look out for them as this would help justify the inferences made in the discussion section. Below is a link of a study that could be of help as you can look out for other studies; Gender-based violence during COVID-19 lockdown: case study of a community in Lagos, Nigeria. African Health Sciences 22(2):79-87 DOI:10.4314/ahs.v22i2.10 https://www.researchgate.net/publication/362674259_Gender-based_violence_during_COVID-19_lockdown_case_study_of_a_community_in_Lagos_Nigeria
--

VERSION 1 – AUTHOR RESPONSE

Reviewer: 1

Dr. N Kumar, Maharishi Markandeshwar Institute of Medical Sciences and Research

Comments to the Author:

The manuscript is very well written and has tried to address one of the major areas especially during the time of COVID-19 Pandemic. There are minor suggestions which if added can further increase the value of your manuscript.

Thank you for your valuable feedback. Below, please find our responses to your comments and suggestions embedded in the manuscript file:

1. Please elaborate the things that were included under sexual harassment (page 1 line 10).

Thank you for this suggestion. We agree that the details of our sexual harassment measure are important for the reader's understanding of the results. Wordcount restrictions in the abstract prevent us from including these details in the abstract, rather we include these details, including wording, operationalization, and source of the measure in the methods section (page 7 lines 35-56).

2. Check for grammar. (page 1 line 19)

Thank you for this comment. We have updated this sentence to read: "Overall, 18.1% of AGYW experienced past-year sexual harassment at the 2020 survey." (Page 1 line 19)

3. It will be very nice if the authors can add the data collection tool/questionnaire with thin the article for better understanding of questions. (Page 7 line 8)

Our data and instruments are publicly available for readers who wish a deeper understanding of the primary outcome as well as other factors assessed.

4. Please mention how patient consent was taken? (Page 7 line 25)

Thank you for this suggestion. We have added language to clarify that participants who had consented for recontact in 2019 were recontacted for the 2020 survey (Page 4 line 32). We have also added details about the consent procedure (Page 5 line 8).

5. It will be better if the authors mention the degree of sexual harassment and violence also, like some may experience extreme degrees, many consider abusive language or threatenings by partner as normal. So, if the authors can include a column for that then it will be very nice. (Page 9 line 25)

Thank you for this suggestion. Unfortunately, we did not assess the severity of harassment experienced by participants, only if they experienced a *change* in occurrence relative to the start of the COVID-19 pandemic. We agree, this would be a valuable contribution to the paper and hope to include such a measure in future data collection. We now acknowledge this as a limitation (p 12, line 20).

6. and by how much it has increased. how many required hospitalization or help. (Page 11 line 22)

Thank you for this suggestion. We do include a measure of changes in occurrence of sexual harassment experiences relative to the COVID-19 pandemic (see page 5, lines 30-32, table 1). Unfortunately, we did not ask participants about the potential health impacts of their sexual harassment experiences, nor help-seeking behaviors related to sexual harassment. We agree, this would be a valuable contribution to the paper and hope to include such a measure in future data collection.

Reviewer: 2

Dr. David Olawade, University of Ibadan

Comments to the Author:

This paper is very concise, comprehensive and articulate. Good knowledge of the study background and sound methodology. The findings are intriguing and the recommendations are insightful, however, there is a need for minor revision as it concerns the discussion section.

There are similar cross-sectional studies carried out in other part of Africa and the world at large that the findings in this study can be compared with. Do look out for them as this would help justify the inferences made in the discussion section. Below is a link of a study that could be of help as you can look out for other studies;

Gender-based violence during COVID-19 lockdown: case study of a community in Lagos, Nigeria.

African Health Sciences 22(2):79-87

DOI:10.4314/ahs.v22i2.10

https://www.researchgate.net/publication/362674259_Gender-based_violence_during_COVID-19_lockdown_case_study_of_a_community_in_Lagos_Nigeria

Thank you for your feedback and for including the recommended paper in your review. We have updated our discussion and limitations sections to include this and other recent cross-sectional studies that examine sexual harassment violence in the African context. (Page 10, line 16; Page 11 line 1; Page 11 line 17; Page 12 lines 15-17; references 38-42)